# Application of Signals with Rippled Spectra as a Training Approach for Speech Intelligibility Improvements in Cochlear Implant Users

**DOI:** 10.3390/jpm12091426

**Published:** 2022-08-31

**Authors:** Dmitry Nechaev, Marina Goykhburg, Alexander Supin, Vigen Bakhshinyan, George Tavartkiladze

**Affiliations:** 1A.N. Severtsov Institute of Ecology and Evolution, 119071 Moscow, Russia; 2National Research Centre for Audiology and Hearing Rehabilitation, 117513 Moscow, Russia; 3Russian Medical Academy of Continuing Professional Education, 125993 Moscow, Russia

**Keywords:** hearing, cochlear implants, rippled spectrum, speech discrimination

## Abstract

In cochlear implant (CI) users, the discrimination of sound signals with rippled spectra correlates with speech discrimination. We suggest that rippled-spectrum signals could be a basis for training CI users to improve speech intelligibility. Fifteen CI users participated in the study. Ten of them used the software for training (the experimental group), and five did not (the control group). Software based on the phase reversal discrimination of rippled spectra was used. The experimental group was also tested for speech discrimination using phonetic material based on polysyllabic balanced speech material. An improvement in the discrimination of the rippled spectrum was observed in all CI users from the experimental group. There was no significant improvement in the control group. The result of the speech discrimination test showed that the percentage of recognized words increased after training in nine out of ten CI users. For five CI users who participated in the training program, the data on word recognition were also obtained earlier (at least eight months before training). The increase in the percentage of recognized words was greater after training compared to the period before training. The results allow the suggestion that sound signals with rippled spectra could be used not only for testing rehabilitation results after CI but also for training CI users to discriminate sounds with complex spectra.

## 1. Introduction

Signals with rippled spectra (rippled signals) are convenient tests for measuring the frequency resolution of hearing. Rippled spectra feature periodically alternating spectral peaks and troughs that form a sort of spectral grid. The resolvable ripple density (the number of ripples per frequency unit) may be considered a convenient measure of the frequency resolution of the auditory system. The resolvable ripple depth may be considered a measure of the spectral modulation resolution. Therefore, sound signals with rippled spectra have been applied to assess the frequency resolving power (FRP) of listeners with normal hearing [1,2], patients with hearing loss [3], and cochlear implant (CI) users [4,5,6]. The FRP for CI users is much lower than that for normal-hearing listeners and features considerable variation.

For ripple pattern resolution measurements, various discrimination tasks have been used in conjunction with the various versions of rippled-spectrum tests: (i) discrimination between ripple noise with constant position of ripples and ripple noise with ripple reversals (the spectral ripple discrimination test) [3,7,8]; (ii) discrimination between a flat and rippled spectrum with varying modulation depth (the spectral ripple detection) [9,10]; (iii) discrimination between the ripple spectrum with drifting ripple phase and constant ripple spectrum (the spectral-temporally modulated ripple test) [11,12,13].

The results of measurements using various rippled-spectrum tests correlated with speech recognition in quiet conditions [3,4,5,6,9,10] as well as in background noise [5,6,8]. The strength of the correlation depended on the test signal parameters and the type of deafness [14].

Rippled spectra tests have been suggested for utilization with respect to clinical goals. These tests may be useful as a nonlinguistic diagnostic tool to estimate the rehabilitation results after cochlear implantation and could predict speech recognition by CI users. Several tests were developed based on various tasks: the spectral discrimination test with constant stimuli [13,15] and the spectral modulation detection task (quick spectral modulation detection and easy quick spectral modulation detection) [9,10,16,17,18,19].

Apart from diagnostic application, rippled signals may be a tool for training CI users to extract information from stimuli generated by the CI. Previously, learning effects on solving ripple discrimination and detection tasks have been investigated [8,12,20,21]. The results were contradictory. For the ripple discrimination test in quiet conditions, there was no learning effect after 12 repeated runs, but a learning effect was observed in noise [8]. Drennan et al. [20] reported that results of a spectral discrimination test remained stable over time. Later, the same research group demonstrated that although spectral-ripple discrimination remained constant over the first year after implantation, 20% of the individuals showed a significant improvement in spectral-ripple discrimination [21]. Drennan et al. [22] reported a learning effect in normal-hearing listeners with CI simulation in noise. De Jong et al. [12] demonstrated a significant learning effect for the spectral-temporally modulated ripple test. The learning effect was observed between 2 and 6 weeks, although an instantaneous learning effect during sessions was not detected.

In the present study, we investigated whether CI users can experience improved ripple spectrum discrimination due to training.

## 2. Materials and Methods

### 2.1. Study Design

Training was performed by multiple repetitions of runs during which the listener had to distinguish between a test and a reference signal that differed from one another by the patterns of spectral ripples, with the ripple density approaching the resolution limit. Apart from the expected training effect, each run provided an estimate of the current ripple density resolution.

### 2.2. Listeners

Fifteen CI users with a diagnosis of bilateral sensorineural hearing loss participated in the study. Data for CI users are summarized in Table 1. All CI users had experience using CI for longer than one year. For all CI users, pure-tonal audiometry in free sound field resulted hearing thresholds of 25–30 dB hearing levels within a frequency range from 0.125 to 8 kHz. The speech development of all CI users corresponded to age norms with no disturbances in the lexical and grammatical structure of speech. 

Ten of the CI users participated in the training procedure (experimental group, EG), and five CI users were in the control group (CG).

### 2.3. Rippled-Spectrum Signals

Rippled-spectrum sound signals were used both for testing ripple density resolution and as a signal for training. The bandwidth of the signals ranged from 0.1 to 8 kHz. Within the frequency band, the spectrum of signals featured several spectral maxima and minima (ripples). The ripple density was frequency proportional; i.e., the ripples looked uniform on a logarithmic frequency scale. The density of ripples was specified in ripples per octave (ripples/oct).

The principle of the ripple phase reversal test was to find the maximum ripple density at which a listener could detect the phase reversals of the spectrum ripples. In the test signal, every 400 ms, the rippled spectrum was replaced with a spectrum of the same parameters except for the opposite position of the spectral peaks and troughs on the frequency scale. The signal contained six segments of alternative ripple phases; thus, the overall signal duration was 2400 ms. The ripple phase in the reference signal was kept constant throughout the signal’s duration. The CI users perceived ripple reversals as a change in the signal timbre if the rippled pattern of the spectrum was resolvable. The highest density of spectral ripples at which the listeners were able to detect phase reversals was considered to be the ripple density’s resolution.

### 2.4. Procedure

For training, ripple-pattern discrimination runs were repeated. In each run, the adaptive variation of the ripple density was performed using a three-alternative force-choice procedure with feedback. Each trial included three stimuli: one with ripples phase reversals (the test) and two with constant ripples phase (the references). The order of the stimuli presentation was varied randomly, trial-by-trial. The task of the CI user was to identify the test signal that differs from the other two. The ripple density varied trial-by-trial adaptively, and a “two-up, one-down” version was used. After two successive correct detections of the test signal, the ripple density in the next trial increased by one step; after every mistake, the ripple density in the next trial decreased by one step. The ripple density varied stepwise using a pseudo-logarithmic scale: 0.7, 1, 1.5, 2, 3, 5, 7, and 10 ripples/oct. Each run continued until 10 turn points (transition from ripple density increase to decrease and back) were obtained. The geometric mean of these 10 points was taken as the threshold estimate for the run.

### 2.5. Speech Discrimination Test

Phonetic material based on polysyllabic balanced speech material was used (30 words in the group). The words contained all phonemes of the Russian language and were pronounced by a male’s voice in Russian. The CI user’s task was to replicate the words. For each group of 30 words, the percentage of correctly replicated words was determined. Correct answers were recorded only if the CI user accurately reproduced all phonemes of the heard word.

Before measurements, CI users had several training sessions intended to make sure that he/she understands the task of replication the words. Different sets of words were used for each test.

### 2.6. Control and Training

Before training, in the clinic, the ripple density resolution and speech discrimination were tested in all CI users in the EG and CG groups. 

For EG, depending on the listeners’ personal circumstances, the training lasted from 4 to 16 weeks (mean 10 weeks), at one run per day. Every run provided an estimate of the current ripple density resolution. The training proceeded at CI users’ home.

After the training, in the clinic, the ripple density resolution and speech discrimination were retested for the EG and CG. For the CG, ripple density resolutions were retested after 16 weeks.

Apart from the speech-discrimination tests performed in the present study, the results of the before-study tests (more than eight months before the training) were available for five CI users of the EG (“long before” data).

### 2.7. Instrumentation

The LabView (National Instruments, Austin, TX, USA) environment was used for software development. 

For the ripple spectrum test, the digitally synthetized signals were digital-to-analog (D/A) converted by a USB 6212 data acquisition board (National Instruments) and played in a free sound field via an SP 90 loudspeaker (Interacoustics, Middelfart, Denmark) located at a distance of 1 m in front of the CI user.

For the speech discrimination test, signals were played with an AC-40 clinical audiometer (Interacoustics, Denmark) and SP 90 loudspeaker (Interacoustics, Denmark) located at a distance of 1 m in front of CI user. The average sound level of the signals was 65 dB SPL in the clinical test.

The training proceeded at CI users’ home using their PC, and digitally synthetized signals were D/A converted by the sound card of PC. During the training, the CI users could use a comfortable sound level by their choice.

### 2.8. Statistical Analysis

All statistical analyses were performed using GraphPad Prism 9 software (GraphPad Software, San Diego, CA, USA).

## 3. Results

In the EG, multiple repetitions of ripple-pattern discrimination runs resulted in an increase in ripple density resolutions (Table 2, Figure 1). The resolution increased from 1.2 times (EG6) to 5.7 times (EG1). Within the EG group, the ripple density resolutions before training and after training significantly differed (*p* = 0.002, Wilcoxon matched pairs test, *N* = 10, *W* = 55). In the CG, the same test did not reveal a significant difference between the ripple density resolution of the test and retest (*p* > 0.9, *N* = 5, *W* = 1).

In majority of the CI users, the individual dynamics of the ripple density resolution demonstrated progressive improvement during training (Figure 2). The regression analysis revealed a significant positive slope of regression lines for all CI users, from 0.015 (ripples/oct)/day (EG6, *p* = 0.029) to 0.094 (ripples/oct)/day (EG3, *p* < 0.001). The only exception was EG9, which featured a negative slope of -0.43 (ripples/oct)/day; this negative slope was not statistically significant (*p* = 0.067).

The results of the word recognition test showed that the percentage of recognized words increased after training in 9 out of 10 EG CI users (Table 3, Figure 3). The Wilcoxon matched-pairs test showed a significant difference between the percentages of recognized words before training and after training (*p* = 0.0039, *N* = 10, *W* = 45). In the CG, the Wilcoxon matched-pairs test did not reveal a significant difference between the percentage of recognized words for the test and retest (*p* > 0.9, *N* = 5, *W* = 2).

We had “long before” (more than 8 months before training) data of word recognition for 5 out of 10 EG CI users. For two of these five CI users (EG 3 and 6), the increase in word recognition was observed only after training and was not observed during the preceding period of using the CI. In one CI user (EG2), the increase in the percentage of word recognition was 10% during 8 months before training and 10% further after 4 months of training. For EG5, the increase in percentage of word recognition was 5% during 18 months before training and 15% after 3 months of training.

It is notable that three patients (EG 1, 5, and 6) reported subjective improvements in speech discrimination after training.

## 4. Discussion

A weak point in the present study was the limited number of CI users. Because of that, EG and CG were rather small, EG and CG were unequal, and CI users have different types of CIs. We assume that there was no device difference that influenced both ripple spectrum resolutions and word recognition. Our assumption was based on the fact that in previous studies, no difference in speech intelligibility was found depending on the manufacturer of the CI system [23]. Lansberger et al. [24] assumed that the ACE (CI 512) processing strategy might in fact benefit spectral resolution by enhancing spectral contrast. The type of array may influence the resolution [14]; however, in the present study, all CI users had a perimodiolar array. This weakness was compensated by a comparison of the control and experimental periods in the same CI users. A comparison of “After training”, “Before training”, and (when available) “Long before” data showed that several (4 to 16) weeks of training for rippled spectra discrimination produced more prominent improvements in speech recognition than the period preceding CI use. Therefore, we suggest that the present study demonstrated significant improvements in both spectral ripple discrimination and speech recognition as a result of training. On average, the spectral ripple discrimination improved by approximately threefold. In previous studies [8,12,20,21], the learning effect has been detected for the spectral temporal modulation task, but those studies did not imply the purpose of training, and there were long intervals between repeated measurements. In the present study, CI users performed the task every day and that might have made the training program more effective.

The data on improvement of ripple pattern discrimination alone do not show that either auditory abilities improved or procedural learning occurred. Procedural learning might appear because EG listeners participated in more sessions than CG listeners. We suppose that the procedural learning for ripple spectra resolutions could not improve word recognition because of substantial differences in testing procedures. Additionally, before the speech test, CI users had training to make sure that they understood the task. We suggest that the improvement in spectral ripple discrimination ability and the percentage of recognized words indirectly indicate perceptual learning in complex spectral discrimination during training.

Based on the obtained results, we hypothesize that spectral ripple discrimination training may be useful for CI users’ learning to resolve separate spectral peaks in complex spectra. This, in turn, can help discriminate and recognize complex sound signals, including speech. Spectral discrimination training could be an addition to analytic and synthetic training approaches.

## Figures and Tables

**Figure 1 jpm-12-01426-f001:**
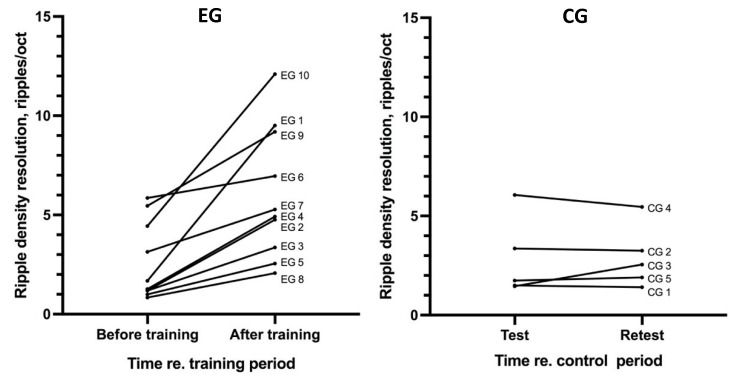
The ripple density resolution (ripples/oct) for the EG (before and after training) and CG (test/retest). Each line indicates the result of one CI user.

**Figure 2 jpm-12-01426-f002:**
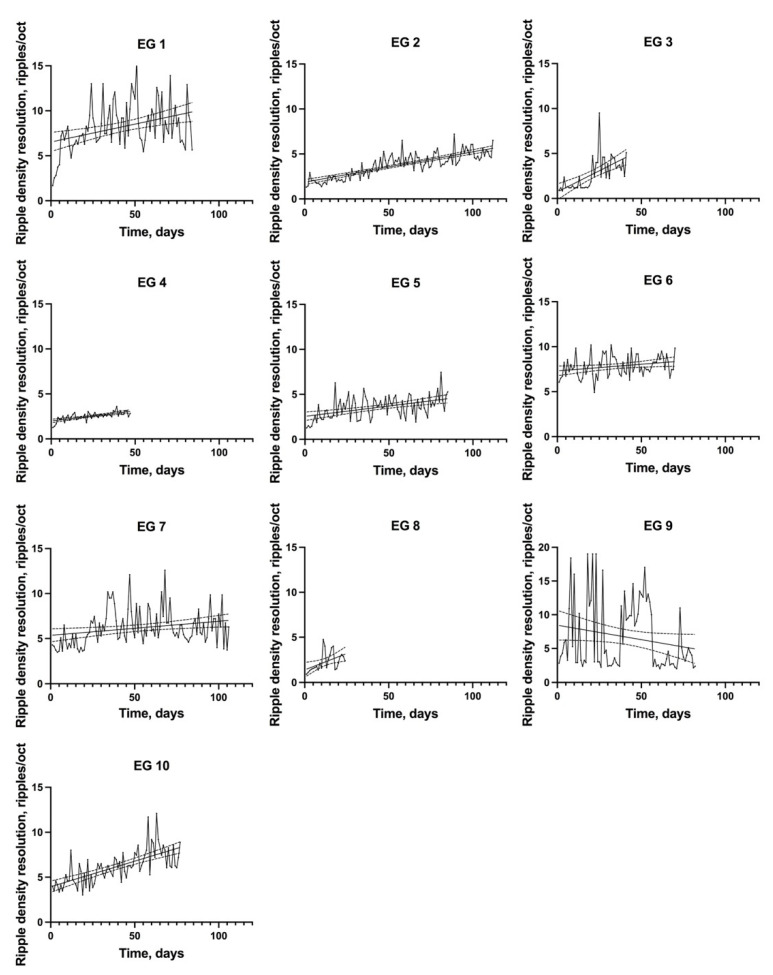
The individual changes of ripple density resolution (ripples/oct) during training. The straight line is a regression line; the dotted lines show 95% confidence bands of the best-fit line.

**Figure 3 jpm-12-01426-f003:**
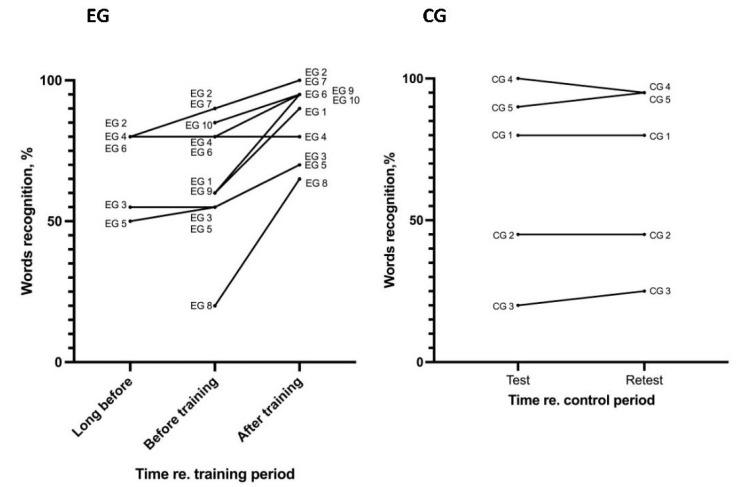
The percentage of recognized words for EG (long before, immediately before training, and after training) and for CG (test/retest). Each line indicates results for one CI user.

**Table 1 jpm-12-01426-t001:** Basic data for the CI users.

ID	Age	CI Model	Implantation Date	Time of Training, Days
EG 1	47	HiRes 90 K Advantage CI MS Electrode	11 December 2019	84
EG 2	61	HiRes 90 K Advantage CI MS Electrode	30 May 2019	112
EG 3	46	HiRes 90 K HiFocus Helix electrode	8 December 2014	41
EG 4	48	HiRes 90 K HiFocus Helix electrode	24 January 2013	48
EG 5	35	HiRes 90 K HiFocus Helix electrode	28 June 2012	84
EG 6	57	CI 512 (CA)	24 October 2018	70
EG 7	37	HiRes 90 K HiFocus Helix electrode	3 April 2012	105
EG 8	28	HiRes 90 K Advantage CI MS Electrode	9 October 2019	28
EG 9	18	HiRes 90 K HiFocus Helix electrode	12 October 2012	82
EG 10	10	HiRes 90 K HiFocus Helix electrode	21 March 2015	77
CG 1	46	CI 512 (CA)	20 June 2017	-
CG 2	33	CI 512 (CA)	6 June 2019	-
CG 3	33	CI 512 (CA)	6 June 2019	-
CG 4	11	Nucleus Freedom CI24RE(CA)	22 June 2011	-
CG 5	46	HiRes 90 K HiFocus Helix electrode	2 November 2011	-

**Table 2 jpm-12-01426-t002:** Ripple density resolution data.

**Experimental Group**			
**Code**	**Before Training** **Ripples/oct**	**After Training** **Ripples/oct**	**After/Before Ratio**
EG1	1.7	9.5	5.6
EG2	1.2	4.8	4.0
EG3	1.2	3.4	2.8
EG4	1.3	4.9	3.8
EG5	1.0	2.6	2.6
EG6	5.9	7.0	1.2
EG7	3.1	5.3	1.7
EG8	0.8	2.1	2.6
EG9	5.4	9.2	1.7
EG10	4.4	12.1	2.8
**Control Group**			
**Code**	**Test**	**Retest**	**Retest/Test Ratio**
CG1	1.5	1.4	0.9
CG2	3.4	3.3	1.0
CG3	1.5	2.6	1.7
CG4	6.1	5.5	0.9
CG5	1.7	1.9	1.1

**Table 3 jpm-12-01426-t003:** Word recognition data.

ID	Long Before, %	Before Training, %	After Training, %
EG1	-	60	90
EG2	80	90	100
EG3	55	55	70
EG4	80	80	80
EG5	50	55	70
EG6	80	80	95
EG7	-	90	100
EG8	-	20	65
EG9	-	60	95
EG10	-	85	95
		Test	Retest
CG1	-	80	80
CG2	-	45	45
CG3	-	20	25
CG4	-	100	95
CG5	-	90	95

## Data Availability

The data that support the findings of this study are available from the corresponding author, Dmitry Nechaev, upon request.

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
