# Peer review of "Application of Signals with Rippled Spectra as a Training Approach for Speech Intelligibility Improvements in Cochlear Implant Users"

_jpm, 2022, doi:10.3390/jpm12091426_

Round 1

Reviewer 1 Report

This study investigated whether CI users performance with rippled stimuli and speech can improve with training with rippled stimuli.

While the goal of this study is important, there were a number of issues with the study. One major issue was that, while the study is set up both in the title and the abstract as investigating the training effects of speech intelligibility, that was not fully investigated in this study.  There appears to be no control group data for the speech intelligibility task and no investigation of the relationship between improvements after training on rippled stimuli and on speech.  This is particularly critical since it appears that such a link is weak or missing, making it difficult to establish a causal relationship between the ripple training and the speech perception improvement.  For example, EG 8 shows a large improvement in speech perception but only a very small improvement in ripple density resolution.

Another major issue is that it is not clear if auditory abilities have actually improved or if only procedural learning has occurred.  It appears that the non-training group only completed the ripple task twice, which may not be sufficient for procedural learning and thus may underlie differences in the training and non-training groups' results.  It would be helpful to show the scores for each training day for the training group to determine if there is a rapid improvement early in training followed by a plateau, suggesting procedural learning.  It would also be helpful to use a different ripple test or spectral test pre-and post-training to verify that there is a generalizable improvement in auditory abilities, particularly in spectral resolution.  There are a lot of other components to speech tests, even if performance on them often correlates with performance on spectral tasks, making it difficult to make a clear case for what is improving with training.

Specific comments:

Line 149: Why were non-parametric tests used?

Line 165-167: The improvement in performance between the two pre-tests, while only occurring for a subset of participants, highlights the issue that procedural learning could play a role in the training effects.

Discussion section: There is largely a difference in the types of CIs include in the training and the control group.  While it is not clear that this would necessarily affect training results, it should be mentioned and discussed.

Minor comments:

Line 24: were obtained -> were also obtained

Line 81: were subjected to the training -> participated in the training

Author Response

  1. “One major issue was that, while the study is set up both in the title and the abstract as investigating the training effects of speech intelligibility, that was not fully investigated in this study. There appears to be no control group data for the speech intelligibility task and no investigation of the relationship between improvements after training on rippled stimuli and on speech.”

The percentage of recognized words for control group is presented (Figure 3, L. 197- 199). With the limited number (15) of CI users available for experiments, control group was as small as 5 listeners. In the revised version, we stress that the small control group and unequal sizes of the experimental and control groups are a weak point of the study (L. 221-231). Therefore, to demonstrate the effect, the pre-training and training periods were added. The data showed that 4 to 16 weeks of training gave more prominent improvement of speech recognition than more than 8 months of the preceding CI use.

  1. “For example, EG 8 shows a large improvement in speech perception but only a very small improvement in ripple density resolution.”

Improvements of speech intelligibility and ripple density resolution cannot be compared directly because they are specified in different units. We do not know, how improvement in spectral resolution converts to improvement of speech recognition because the speech recognition depends on a variety of factors. In particular, the word recognition can never exceed 100%.

Then, in EG8 the ripple pattern resolution improved by 2.6 times (Table 2). This improvement is not VERY small.

Because of inter-individual variation of both ripple density resolution (Table 2) and speech recognition (Table 3), the training effect was assessed statistically. The effects on both ripple density resolution and word recognition were statistically significant (ll. 170-172 and 195-197).

“Another major issue is that it is not clear if auditory abilities have actually improved or if only procedural learning has occurred.  It appears that the non-training group only completed the ripple task twice, which may not be sufficient for procedural learning and thus may underlie differences in the training and non-training groups' results…“

The training affected both the ripple density resolution and word recognition. The word recognition was measured by a procedure that was quite different from that for the ripple resolution, so the procedural learning could not help for word recognition. This result allowed us to suggest that the training improves the capability to process the CI information (L. 240-248).

“…It would be helpful to show the scores for each training day for the training group to determine if there is a rapid improvement early in training followed by a plateau, suggesting procedural learning.”

Thank you. Individual dynamics of ripple density resolution is presented (L. 185-189, Figure 2).

“It would also be helpful to use a different ripple test or spectral test pre-and post-training to verify that there is a generalizable improvement in auditory abilities, particularly in spectral resolution.  There are a lot of other components to speech tests, even if performance on them often correlates with performance on spectral tasks, making it difficult to make a clear case for what is improving with training.”

The mechanism of improvement of speech discrimination due to the training of ripple pattern discrimination is not revealed by the present study. The use a different ripple test hardly could provide this information because of a substantial difference between rippled signals and speech.

“Line 149: Why were non-parametric tests used?”

For small groups, the non-parametric statistic is more adequate than the parametric statistic. It may reveal a statistical significance of a trend that cannot be demonstrated by the parametric statistics in small groups.

“Line 165-167: The improvement in performance between the two pre-tests, while only occurring for a subset of participants, highlights the issue that procedural learning could play a role in the training effects.”

For speech discrimination tests, CI users had several training sessions intended to make sure that he/she understands the task of replication the words, and the different sets of words were used for each test (L. 135 – 137). So, their ability for understanding the task could not substantially differ in “long before”, “before training”, and “after training” tests.

“Discussion section: There is largely a difference in the types of CIs include in the training and the control group.  While it is not clear that this would necessarily affect training results, it should be mentioned and discussed.”

Thank you (L. 223-227)

Minor comments:

“Line 24: were obtained -> were also obtained”

Thank you (L. 24)

“Line 81: were subjected to the training -> participated in the training”

Thank you (L. 90)

Reviewer 2 Report

Introduction

The introduction should include an explanation of the basic issues that the author writes about, such as what are signals with rippled spectra?, what are their characteristics?, how do they differ from others?, why exactly this way and this method were chosen for training?

Another aspect to consider is explaining to the reader why CI patients need auditory training and how (the classical) auditory training for CI patients is carried out.

An important question that arises from reading the papers is: what can affect the discrimination and better understanding of speech in patients with CI

Materials and Methods

In my opinion, the author should provide the results of the hearing test (tonal audiometry and level of speech understanding). The ability to understand speech and its level is crucial in determining later results

Speech discrimination test- for what intensity was the test conducted, was the opposite ear masked, what was the azimuth?

In the discussion there is no reference to other publications, the author does not present and compare the results with other works

The study group is very small, some patients do not have complete information

one CI user (EG2), the increase was 10% for 8 months
before training and 10% after just 4 months of training”– increase is the same?

Author Response

“The introduction should include an explanation of the basic issues that the author writes about, such as what are signals with rippled spectra? what are their characteristics?, how do they differ from others?, why exactly this way and this method were chosen for training?”

Thank you. In the revised version, Introduction starts with a short characteristic of rippled signals (L. 32-37).

“Another aspect to consider is explaining to the reader why CI patients need auditory training and how (the classical) auditory training for CI patients is carried out.”

Thank you. Training may help CI users to extract information from stimuli generated by CIs (L. 60-61).

“An important question that arises from reading the papers is: what can affect the discrimination and better understanding of speech in patients with CI”

The presented data do not reveal the mechanisms of improving the speech recognition. They just pretend to demonstrate the improvement. This is a necessary step before investigating the mechanisms.

Materials and Methods

“In my opinion, the author should provide the results of the hearing test (tonal audiometry and level of speech understanding). The ability to understand speech and its level is crucial in determining later results.”

Thank you. In the revised version, both the tonal audiometry data and speech recognition data are presented (L. 86-88 and Table 3).

“Speech discrimination test- for what intensity was the test conducted, was the opposite ear masked, what was the azimuth?”

The level of the speech tests is presented (L. 159-160). The masker was not used because CI users participated in the study have bilateral deafness (l. 81).

“In the discussion there is no reference to other publications, the author does not present and compare the results with other works”

We tried to do all our best for collecting the literature data. Which publications are missed?

“The study group is very small, some patients do not have complete information”

Yes, the available group of CI users was limited. The available information is presented in Table 1.

“one CI user (EG2), the increase was 10% for 8 months before training and 10% after just 4 months of training”– increase is the same?”

In the revised version, we reformulated this part (211 -217), and added the table 3 with words recognition results.

Round 2

Reviewer 1 Report

The manuscript has been improved with this revision, and the addition of the control group for speech testing provides good support for the authors’ claims.

Specific comments:

Line 36-37: I would argue that loudness sensitivity could also play a role

Line 66-68: This sentence seems to contradict itself.  Is a group effect being contrasted with an individual effect?  This should be made clear.

Minor comments:

Line 26: allow to suggest -> suggest

Line 87: resulted -> demonstrated

Line 90: users were -> users

Line 99: as signal -> as a signal

Line 136: The different -> Different

Line 186: training Figure -> training, Figure

Line 226: should “assumed” by “suggested”?

Line 233: than more than -> than the

Line 240: day that -> day and that

Line 241: might made -> might have made

Line 242: show, -> show whether

Line 243: learning, occurred -> learning occurred

Line 243: procedure -> procedural

Line 247: before speech -> before the speech

Author Response

Dear Reviewer!

We are grateful to you for the great work you have done to improve our paper. 

Below are our answers,

Specific comments:

Line 36-37: I would argue that loudness sensitivity could also play a role

Loudness plays a role for ripple-pattern resolution. However, the specified and the preceding sentences state which sound parameters might be measured by rippled signals. For measuring loudness sensitivity, rippled signals ate not commonly used.

Line 66-68: This sentence seems to contradict itself.  Is a group effect being contrasted with an individual effect?  This should be made clear.

The sentence is modified (ll. 67-68).

Minor comments:

Line 26: allow to suggest -> suggest

Thank you (l. 26)

Line 87: resulted -> demonstrated

Thank you (l. 86)

Line 90: users were -> users

Thank you (l. 89)

Line 99: as signal -> as a signal

Thank you (l. 98)

Line 136: The different -> Different

Thank you (l. 135)

Line 186: training Figure -> training, Figure

Thank you (l. 185)

Line 226: should “assumed” by “suggested”?

Thank you (l. 225)

Line 233: than more than -> than the

Thank you (l. 232)

Line 240: day that -> day and that

Thank you (l. 239)

Line 241: might made -> might have made

Thank you (l. 240)

Line 242: show, -> show whether

Thank you (ll. 241-242)

Line 243: learning, occurred -> learning occurred

Thank you (l. 242)

Line 243: procedure -> procedural

Thank you (l. 242)

Line 247: before speech -> before the speech

Thank you (l. 246)
